# Synthesis of magnesium-nitrogen salts of polynitrogen anions

Dominique Laniel [1]*, Bjoern Winkler[2], Egor Koemets[3], Timofey Fedotenko[1], Maxim Bykov [3], Elena Bykova [4], Leonid Dubrovinsky [3] & Natalia Dubrovinskaia [1]

The synthesis of polynitrogen compounds is of fundamental importance due to their potential as environmentally-friendly high energy density materials. Attesting to the intrinsic difficulties related to their formation, only three polynitrogen ions, bulk stabilized as salts, are known. Here, magnesium and molecular nitrogen are compressed to about 50 GPa and laser-heated, producing two chemically simple salts of polynitrogen anions, $MgN_4$ and $Mg_2N_4$. Single-crystal X-ray diffraction reveals infinite anionic polythiazyl-like 1D N-N chains in the crystal structure of $MgN_4$ and *cis*-tetranitrogen $N_4^{4-}$ units in the two isosymmetric polymorphs of $Mg_2N_4$. The *cis*-tetranitrogen units are found to be recoverable at atmospheric pressure. Our results respond to the quest for polynitrogen entities stable at ambient conditions, reveal the potential of employing high pressures in their synthesis and enrich the nitrogen chemistry through the discovery of other nitrogen species, which provides further possibilities to design improved polynitrogen arrangements.

[1] Material Physics and Technology at Extreme Conditions, Laboratory of Crystallography, University of Bayreuth, 95440 Bayreuth, Germany. [2] Institut für Geowissenschaften, Abteilung Kristallographie, Johann Wolfgang Goethe-Universität Frankfurt, Altenhöferallee 1, D-60438 Frankfurt am Main, Germany. [3] Bayerisches Geoinstitut, University of Bayreuth, 95440 Bayreuth, Germany. [4] Photon Science, Deutsches Elektronen-Synchrotron, Notkestrasse 85, 22607 Hamburg, Germany. *email: dominique.laniel@uni-bayreuth.de

Polynitrogen entities are recognized to be ideal constituents of high energy density materials (HEDM) on account of the tremendous energy released upon the conversion of singly or doubly bonded N–N atoms to triple-bonded molecular nitrogen[1]. As a direct consequence of their energetic potential, these low order bonds are highly unstable. So far, only three homonuclear polynitrogen species have been bulk stabilized at ambient conditions: $N_3^-$, $N_5^+$, and $N_5^{-2}$. In particular, the stabilization and isolation of the $N_5^+$ and $N_5^-$ ions is arduous and typically achieved in salts with a complex, multi-elements, counter ion[3–8]. Other polynitrogen units are known to exist, such as $N_4^+$ and $N_4$, but they are deemed unstable as they have a lifetime below the millisecond threshold[9] and can only be kept intact by being trapped in inert matrices at ultralow temperatures[10,11].

The application of pressure opens up unique possibilities to explore and produce polynitrogen entities. Indeed, pressure generally promotes electronic delocalization and thus favors the formation of extended and polymeric networks. As a matter of fact, the only bulk stable neutral energetic polynitrogen specie, cubic-gauche polymeric nitrogen (cg-N), was synthesized under pressure, at 110 GPa[12]. In that same pressure regime, other polynitrogen anions were discovered, such as infinite 1D armchair chains (in $FeN_4$ and $ReN_8 \cdot xN_2$)[13,14] and the pentazolate ring ($N_5^-$), found in $CsN_5$[15]. Despite numerous attempts, none of these compounds could be recovered at ambient conditions. The sole exception is the $(Li)^+(N_5)^-$ salt, produced near 50 GPa and retrieved at ambient conditions[16]. Unlike the vast majority of poly-N ions formed through classical chemistry methods, their pressure-formed counterparts have the significant advantage of being extremely simple, typically with a single element acting as the counter ion.

Atomistic model calculations highlight the outstanding potential of high pressure experiments, as a plethora of polynitrogen entities have been predicted to be stable[17–19]. Among them, the compression of magnesium and nitrogen was predicted to produce the $MgN_3$, $MgN_4$, and $MgN_{10}$ salts comprised of exotic anionic benzene-like $N_6$ rings, infinite 1D armchair chains and pentazolates, respectively[17–19]. To the best of our knowledge, these predictions have not been tested until now.

Here, we demonstrate that the $(Mg)^{2+}(N_4)^{2-}$ and the $(Mg_2)^{4+}(N_4)^{4-}$ salts are synthesized by compressing and laser-heating magnesium and molecular nitrogen samples above 50 GPa. Single-crystal X-ray diffraction measurements establish polynitrogen entities, namely infinite anionic polythiazyl-like 1D N-N chains and cis-tetranitrogen $N_4^{4-}$ species, to compose the $MgN_4$ and $Mg_2N_4$ solids, respectively. Upon the full pressure release, the $\beta$-$Mg_2N_4$ compound undergoes an isosymmetric phase transition into the $\alpha$-$Mg_2N_4$ salt, also comprised of cis-tetranitrogen $N_4^{4-}$ units. These results demonstrate the recoverability to ambient conditions of the pressure-produced $N_4^{4-}$ entity, emphasizing the potential and importance of the high pressure approach for the discovery and synthesis of improved polynitrogen species.

## Results and discussion

**Synthesis and characterization of the Mg-N compounds.** With the goal of synthesizing nitrogen-rich Mg-N solids, we compressed micrometer-size pure magnesium pieces surrounded by a large volume of molecular nitrogen—used as both a pressure transmitting medium and a reagent—up to about 60 GPa in four diamond anvil cells (DACs). To facilitate a chemical reaction, samples were laser-heated using YAG lasers at pressures of 28.0, 33.0, 43.4, 52.2, 52.4, 52.7, and 58.1 GPa. The specific pressure-temperature paths followed for all samples are summarized in Supplementary Table 1. Metallic Mg served as a YAG laser absorber. Laser-heating at pressures below 52.2 GPa resulted in the formation of a previously known compound, $Mg_3N_2$ (space group $C2/m$),[20] identified by both X-ray diffraction and Raman spectroscopy measurements (see Supplementary Figures 1 and 2 as well as Supplementary Table 2). Above the threshold pressure of 52.2 GPa, heating the samples to at least 1850 K led to the growth of the Mg piece—supposedly due to nitrogen diffusing into it—, the appearance of two sets of Raman modes and diffraction lines which did not correspond to known phases of either $Mg_3N_2$[20], pure Mg[21], or pure $N_2$[22,23] (Supplementary Figs. 4–7). Single-crystal X-ray diffraction revealed the crystal structures of two compounds with compositions $Mg_2N_4$ and $MgN_4$ (crystallographic data are given in Supplementary Table 3 and Supplementary Table 4).

The $MgN_4$ compound has an orthorhombic structure (Ibam space group) with the lattice parameters $a = 3.5860(13)$ Å, $b = 7.526(3)$ Å and $c = 5.1098(17)$ Å at 58.5 GPa (see Table 1). The magnesium atoms are eight-fold coordinated by nitrogen atoms (see Fig. 1), which are arranged in exotic planar infinite zigzag

## Table 1 Crystallographic data for the MgN₄, β-Mg₂N₄ and α-Mg₂N₄ compounds

|  | $MgN_4$ | $\beta$-$Mg_2N_4$ | $\alpha$-$Mg_2N_4$ |
|---|---|---|---|
| Pressure (GPa) | 58.5 | 58.5 | 0.0001 (1 bar) |
| Space group | Ibam | $P2_1/n$ | $P2_1/n$ |
| $a$ (Å) | 3.5860(13) | 7.113(5) | 7.5182(9) |
| $b$ (Å) | 7.526(3) | 5.828(6) | 6.5426(11) |
| $c$ (Å) | 5.1098(17) | 8.800(9) | 13.4431(19) |
| $\beta$ (°) | 90 | 104.00(7) | 130.080(17) |
| $V$ (Å³) | 137.90(9) | 354.0(6) | 505.95(18) |
| Fractional atomic coordinates (x; y; z) | Mg: (0; 0; 0.25) | Mg1: (0.49079; 0.7820;0.36972) | Mg1: (−0.03444; −0.26972; −0.14194) |
|  | N1: (0.6584; 0.83344;0.5) | Mg2: (0.87494; 1.0122;0.65121) | Mg2: (0.19484; −0.24500; −0.43089) |
|  | N2: (0.2928; 0.3059; 0.5) | Mg3: (0.27163; −0.0478; 0.08387) | Mg3: (−0.15418; −0.47368; −0.39088) |
|  |  | Mg4: (0.73162; 1.0462; 0.26126) | Mg4: (−0.05204; 0.03340; −0.31550) |
|  |  | N1: (0.9146; 0.7621; 0.4030) | N1: (0.2053; 0.2398; −0.29135) |
|  |  | N2: (0.5752; 0.2424; 0.4128) | N2: (0.0799; −0.2324; −0.32480) |
|  |  | N3: (0.5700; 0.4601; 0.3759) | N3: (0.1944; −0.0551; −0.10797) |
|  |  | N4: (1.0672; 0.6905; 0.5081) | N4: (−0.3561; −0.1649; −0.32605) |
|  |  | N5: (1.0586; 0.7188; 0.6557) | N5: (−0.0018; −0.5637; −0.18680) |
|  |  | N6: (0.2820; 0.2132; 0.2432) | N6: (0.1442; 0.3256; −0.39668) |
|  |  | N7: (0.4332; 0.1114; 0.3403) | N7: (−0.3255; 0.2339; −0.45463) |
|  |  | N8: (0.7714; 0.8544; 0.4546) | N8: (−0.4397; −0.2489; −0.43838) |

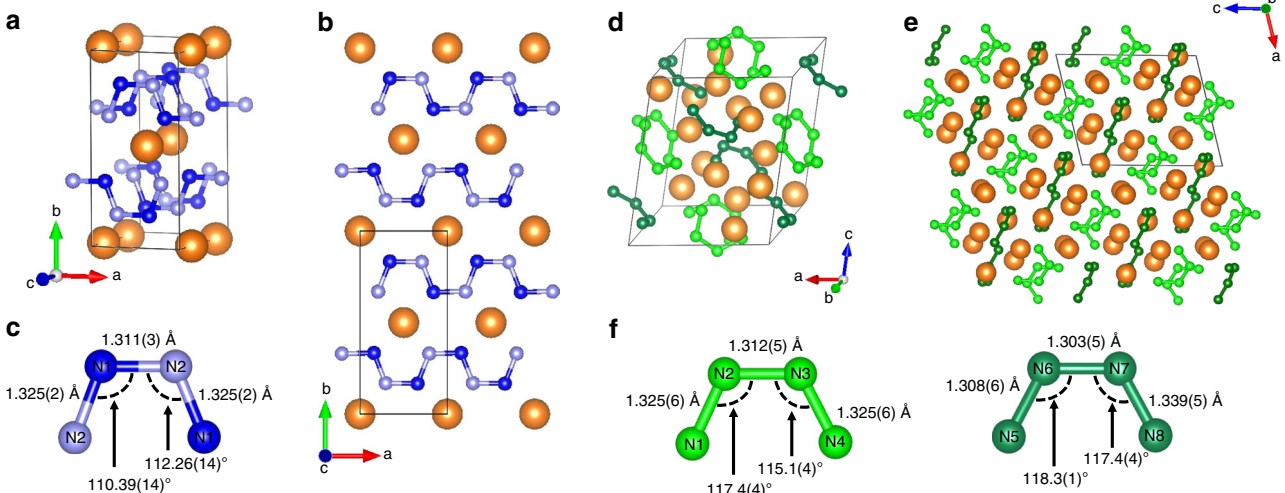

**Fig. 1** The crystal structure of the MgN$_4$ and $\beta$-Mg$_2$N$_4$ salts at 58.5 GPa. **a** The unit cell of MgN$_4$ (the light blue, dark blue and orange spheres represent the N1, N2 and Mg atoms, respectively); **b** a projection of the MgN$_4$ structure along the $c$-axis, emphasising 1D chains of nitrogen atoms aligned along the $a$-axis; **c** a repeating N$_4^{2-}$ subunit of a chain with the N-N distances and angles indicated; **d** the unit cell of $\beta$-Mg$_2$N$_4$ (the light green and dark green spheres represent the four distinct nitrogen atoms forming the $a$-N$_4^{4-}$ and $b$-N$_4^{4-}$ units, respectively, the orange spheres represent Mg atoms); **e** a projection of the $\beta$-Mg$_2$N$_4$ structure along the $b$-axis allowing to see the alternating layers of isolated $a$-N$_4^{4-}$ and $b$-N$_4^{4-}$ units, intercalated with Mg$^{2+}$ ions. **f** The $a$-N$_4^{4-}$ (left) and $b$-N$_4^{4-}$ (right) entities with bond lengths and angles indicated

N-N chains parallel to the $a$-axis, akin to those found in FeN$_4$ and ReN$_8$·$x$N$_2$ near 110 GPa[13,14]. While the MgN$_4$ compound's crystal chemistry matches the theoretical calculations[17–19], the measured lattice parameters do not, as the $c$ parameter is double the predicted value and the structure adopts an $I$-type unit cell, opposite to the calculated $C$-type (see Supplementary Discussion). As shown in Supplementary Fig. 18, by drawing the reciprocal lattice corresponding to the predicted parameter $c$ ($c = 2.5549(17)$ Å), reflections appear at the mid-point of the lattice vector $c^*$; weak but distinctly visible. This means that the $c$ value is actually two times larger ($c = 5.1098(17)$ Å) than the predicted one. The $C$-centering is contradicted by more than 130 experimentally observed reflections. Our DFT calculations reveal the $Ibam$ MgN$_4$ structure to have an enthalpy of about 10 kJ/mol lower than the predicted $Cmmm$ MgN$_4$ solid and to be dynamically stable at 50 GPa (see Supplementary Fig. 10). The full experimental crystallographic data and Raman spectra of $Ibam$ MgN$_4$ are presented in detail in the Supplementary Table 3 and the Supplementary Fig. 5, along with their further comparison to the theoretically computed data (see Supplementary Fig. 8)[17,18].

The structure of the high pressure Mg$_2$N$_4$ phase (Fig. 1) has a monoclinic $P2_1/n$ symmetry with lattice parameters $a = 7.114(3)$ Å, $b = 5.824(2)$ Å, $c = 8.804(4)$ Å and $\beta = 104.04(3)°$ at 58.5 GPa (see Table 1). We name it $\beta$-Mg$_2$N$_4$, to distinguish from its ambient pressure modification $\alpha$-Mg$_2$N$_4$ (see below). The unit cell of $\beta$-Mg$_2$N$_4$ contains twelve symmetrically independent atoms, four Mg and eight N. The atomic arrangement gives rise to an exotic poly-N entity: a N$_4$ unit with a formal charge of 4−. The tetranitrogen anion has a surprising $cis$-like shape, analogous to the nitrogen skeleton in $cis$-tetrazene—here observed as an isolated molecule[24,25]. The eight nitrogen atoms form two discernable N$_4^{4-}$ units, hereafter named $a$-N$_4^{4-}$ and $b$-N$_4^{4-}$, with slight variations in their N-N distances. At 58.5 GPa, $a$-N$_4^{4-}$ has bond lengths of 1.325(6), 1.312(5) and 1.325(6) Å while $b$-N$_4^{4-}$ has 1.308(6), 1.303(5), and 1.339(5) Å, for the first edge bond, the center bond, and the second edge bond, respectively. These modest differences underline a subtle but complex dissimilarity in their chemical environment, namely their proximity and coordination with the Mg$^{2+}$ ions.

**Theoretical insight on the $\beta$-Mg$_2$N$_4$ salt.** The DFT model calculations confirm the $\beta$-Mg$_2$N$_4$ structural model in great detail and hence can confidently be employed to analyze its electronic structure (see Supplementary Table 5 for the complete analysis). The inspection of the total and partial density of states of $\beta$-Mg$_2$N$_4$ unambiguously exposes that there is only a very weak covalent bonding between N$_4^{4-}$ entities, and electron difference maps exhibits no charge accumulation between the Mg and N atoms. Instead, a strong ionic interaction between the Mg and N ions is observed: a Mulliken analysis shows that Mg is essentially in its charged formal state and a semi-quantitative comparison to MgCl$_2$ implies that the bonding between Mg$^{2+}$ and the N$_4^{4-}$ entities is even slightly more ionic than between Mg$^{2+}$ and Cl$^-$. Within the N$_4^{4-}$ entities, the edge and center bonds have the same Mulliken bond population, consistent with their similar bond lengths (~1.318 ± 0.015 Å). This theoretical insight exhibits the stark differences in electronic density configuration between the N$_4^{4-}$ entities in $\beta$-Mg$_2$N$_4$ and those previously observed in $trans$-tetrazene or tetrazadiene complexes, in which the N$_4^{4-}$ units are instead stabilized through strong covalent bonds and bond length disparities denote the single and double bond character of the edge and center bonds, respectively[24,25].

As the $P2_1/n$ $\beta$-Mg$_2$N$_4$ compound—and its crystal chemistry—had not been predicted by previous theoretical calculations, its enthalpy at 50 GPa was compared to the enthalpies of the $P6_3/mcm$ and the $Cmcm$ MgN$_2$ structures that had been predicted[17,18]. We found that $P2_1/n$ $\beta$-Mg$_2$N$_4$ is energetically competitive with the two predicted structure as all were found to be equal within 5 kJ/mol, which is less than the uncertainty of our calculations. The Raman spectrum of $P2_1/n$ $\beta$-Mg$_2$N$_4$ was also computed and reproduces well the experimental spectrum recorded at 49.7 GPa, as seen in Supplementary Fig. 11.

**Recovery of the $\alpha$-Mg$_2$N$_4$ salt to ambient conditions.** After the synthesis of the $\beta$-Mg$_2$N$_4$ compound, the DACs were slowly decompressed and Raman as well as XRD measurements were performed at each step of the decompression to track possible changes. Both the vibrational modes (Fig. 2) and the lattice

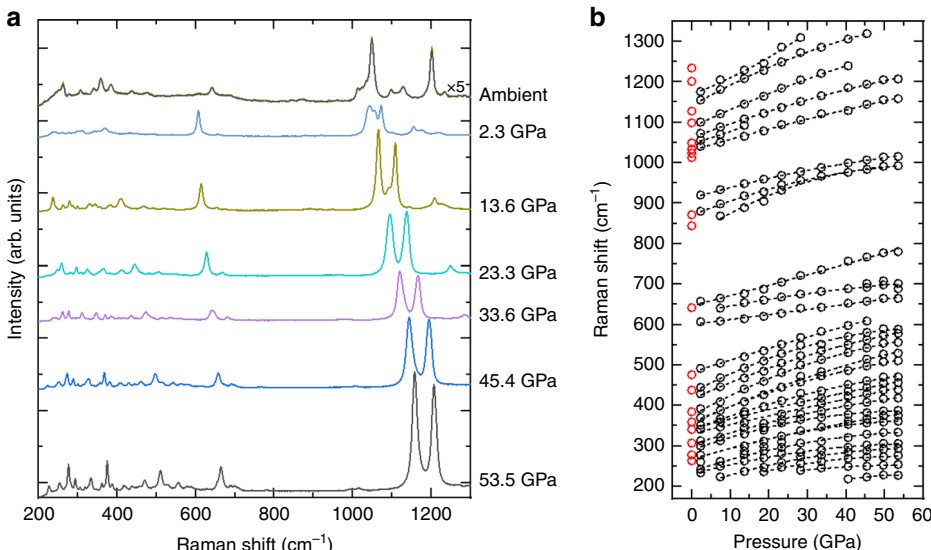

**Fig. 2** Evolution of the Raman modes of the $Mg_2N_4$ compounds with pressure. **a** Typical Raman spectra of $Mg_2N_4$ taken during decompression from 53.5 GPa to ambient conditions. The spectra at ambient conditions is markedly different than those at higher pressures, evidencing a phase transition. **b** Pressure dependence of the Raman modes' frequencies, which evolve smoothly and continuously down to 2.3 GPa. Red dots indicate the modes at ambient pressure. The spectra are offset along the $y$-axis for clarity and ×5 indicates that the spectrum is five-times magnified

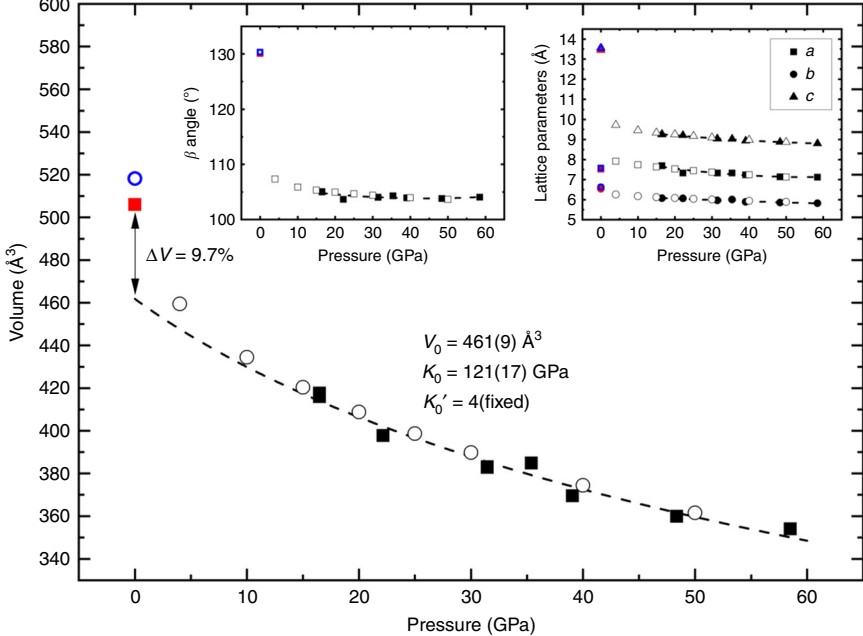

**Fig. 3** Unit cell volume of the $Mg_2N_4$ compounds as a function of pressure. All solid and open symbols are experimental and theoretical data points, respectively, while the dashed line is the fit of the experimental PV data of the $\beta$-$Mg_2N_4$ compound using the second order Birch-Murnaghan equation of state (BM2 EoS) ($V_0 = 461(9)$ Å$^3$ and $K_0 = 121(17)$ GPa). Red and blue symbols are the experimental and theoretical, respectively, unit cell volume of $\alpha$-$Mg_2N_4$ at ambient pressure. The extrapolation of the experimental equation of state suggests a volume jump of 9.7% between the high pressure $\beta$-$Mg_2N_4$ and the ambient pressure $\alpha$-$Mg_2N_4$ phases. The fit of the theoretical PV data using the BM2 EoS gives for $\beta$-$Mg_2N_4$ $V_0 = 470.23$ Å$^3$ and $K_0 = 110.83$ GPa. The theoretical volume difference is thus of 10.2% between the calculated $\beta$-$Mg_2N_4$ and $\alpha$-$Mg_2N_4$. The insets show the dependence of the unit cell parameters on pressure. The full (open) black and full red (open blue) symbols represent experimental (theoretical) data from the $\beta$ and $\alpha$ phases of $Mg_2N_4$. The slightly higher volume obtained from the DFT calculations, compared to the experimental values, shows the underbinding in GGA[26]

parameters (Fig. 3 and Supplementary Tables 5–8) vary smoothly with pressure, displaying no sign of a chemical reaction, decomposition or phase transition down to 2.3 GPa. However, the opening of the DACs in air leads to the loss of the Raman signal from the sample, caused by its deterioration, presumably following a chemical reaction with the water or the oxygen in air (see Supplementary Fig. 13). Therefore, at various pressures

below 18 GPa, the DACs were opened in a glovebox under an inert atmosphere of argon. After the release of pressure, the DACs were closed, still in the glovebox, so that the samples were preserved in an inert atmosphere at ambient pressure. Micro-photographs of a sample before and after the complete release of pressure are shown in Supplementary Fig. 12. The Raman spectra of these decompressed samples appeared to be different from

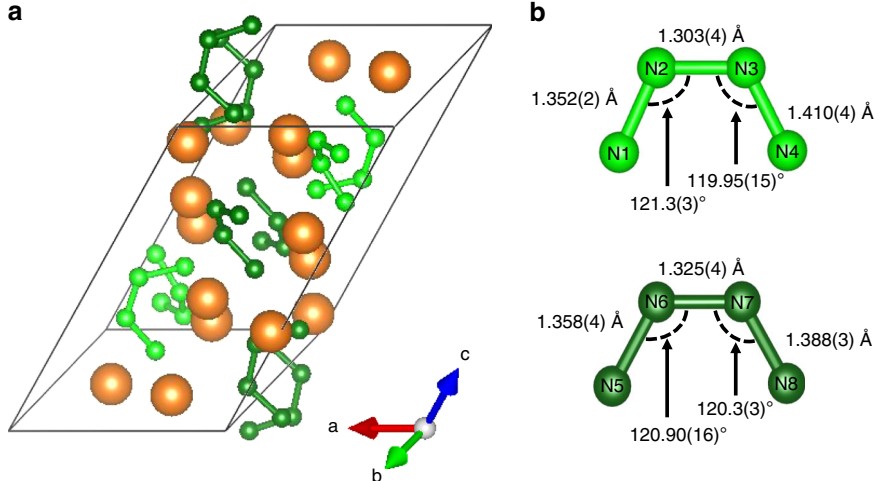

**Fig. 4** Crystal structure of the $\alpha$-Mg$_2$N$_4$ salt at ambient conditions. **a** The unit cell of $\alpha$-Mg$_2$N$_4$ (see also Supplementary Fig. 16). **b** The bond lengths and angles in the two distinct N$_4^{4-}$ entities: the $a'$-N$_4^{4-}$ (light green, top) and $b'$-N$_4^{4-}$ (dark green, bottom). The orange and green spheres represent Mg and N atoms, respectively

those previously observed for $\beta$-Mg$_2$N$_4$. To clarify their structure and chemical composition, the samples were investigated employing single-crystal and powder XRD.

The single-crystal XRD analysis revealed a polymorph of Mg$_2$N$_4$ ($\alpha$-Mg$_2$N$_4$), which still contains two distinct N$_4^{4-}$ units (Fig. 4). Its structural details are found in Table 1 while the full crystallographic data are provided in the Supplementary Table 9. A Le Bail refinement performed on a diffractogram collected from a $\alpha$-Mg$_2$N$_4$ sample at ambient conditions (i.e., after the release of gaseous N$_2$) revealed no other phases, as seen in Supplementary Fig. 15. The structure has the same monoclinic symmetry ($P2_1/n$ space group) with lattice parameters of $a = 7.5182(9)$ Å, $b = 6.5426(11)$ Å, $c = 13.4431(19)$ Å, and $\beta = 130.080(17)°$. The unit cell is strongly deformed in comparison to the high pressure polymorph, with a large increase of the unit cell parameters $c$ and $\beta$ as well as a decrease of the unit cell parameter $a$. The stacking of the N$_4^{4-}$ units in $\alpha$-Mg$_2$N$_4$ also differs from that in the high pressure $\beta$-Mg$_2$N$_4$ phase, as shown in Supplementary Fig. 17. The structural model is again confirmed in great detail by DFT calculations, which also show that the $\alpha$-Mg$_2$N$_4$ is more stable than the $\beta$-Mg$_2$N$_4$ by 10 kJ/mol at ambient conditions (see Supplementary Discussion).

In the $\alpha$-Mg$_2$N$_4$ phase, both of the N$_4^{4-}$ entities display slightly longer N-N distances. In $a'$-N$_4^{4-}$, the bond lengths are of 1.352(2), 1.303(4) and 1.410(4) Å, while in $b'$-N$_4^{4-}$ they are 1.358(4), 1.325(4) and 1.388(3) Å, for the first edge bond, the center bond and the second edge bond, respectively. This resembles the bonds lengths measured in some tetrazadiene complexes (namely [Ir(RNNNNR) (CO)(PPh$_3$)$_2$][BF$_4$], where R is 4-FC$_6$H$_4$)[25]. The DFT calculations show that there are essentially no differences in the character of chemical bonding between the Mg$^{2+}$ and the N$_4^{4-}$ entities in $\alpha$-Mg$_2$N$_4$ and $\beta$-Mg$_2$N$_4$, implying that it is still strongly ionic. The energy released upon the decomposition of $\alpha$-Mg$_2$N$_4$ into Mg$_3$N$_2$ and molecular N$_2$ is calculated to be about 1.9 kJ/g, which is about two times less than known for TNT[27,28].

The $\alpha$-Mg$_2$N$_4$ salt encapsulated in inert atmosphere persisted at ambient conditions for several months, as evidenced by single-crystal X-ray diffraction and Raman spectroscopy (see Supplementary Fig. 14). Although Raman spectroscopy measurements suggest the stability of the MgN$_4$ compound at least down to 0.9 GPa (see Supplementary Fig. 9), it was not detected by XRD at a pressure other than 58.5 GPa. The DFT calculations suggest this

compound to be elastically unstable at ambient conditions, as the $c_{44}$ and $c_{66}$ parameters were calculated to have values below 0 GPa. Further experiments and calculations are underway to conclusively determine the stability domain of MgN$_4$.

The high pressure investigation of the Mg-N system unveiled an exotic and unexpected chemistry. First, a compound featuring infinite 1D polynitrogen chains was obtained near 50 GPa—half of the pressure previously thought to be required[13,14]. Second, in the same pressure domain, previously unknown (N$_4$)$^{4-}$ poly-nitrogen anions were synthesized and stabilized in the simple Mg$_2$N$_4$ salt. Its high pressure polymorph ($\beta$-Mg$_2$N$_4$) undergoes a phase transition on pressure release, but the (N$_4$)$^{4-}$ polynitrogen units persist in its low pressure polymorph ($\alpha$-Mg$_2$N$_4$), adding a unique (N$_4$)$^{4-}$ anion to a yet very short list of poly-N entities bulk stabilized at ambient conditions. The tetranitrogen anion can spark further research for producing improved energetic polynitrogen compounds. Moreover, these results underline the potential and possibilities that are enabled by high pressure chemistry, along with its applicability to produce compounds compatible with ambient conditions.

## Methods

**Sample preparation**. A few pure magnesium flakes of typically about $10 \times 10 \times 5$ μm$^3$ in size were positioned on one of the diamond anvils, with culet diameter of 250 μm. Rhenium was used as the gasket material. A small ruby chip was loaded along with the sample and used to determine the pressure inside the pressure chamber[29]. The diamond anvil cell (DAC) pressure chamber was then loaded with pure molecular nitrogen (~1200 bars) in a high pressure vessel. Pure nitrogen, acting as both a pressure transmitting medium and a reagent, was always largely in excess with respect to magnesium.

Throughout the procedure, care was taken to minimize the exposure of Mg to air. On average, the magnesium flakes were exposed for about 20 min to air during the complete loading procedure. According to previous investigations[30], an exposure of this time span is expected to produce a 20–50 nm protective oxide film at the surface of Mg, which prevents further chemical reaction between Mg and elements in the air. Considering the volume of the loaded Mg pieces (about $10 \times 10 \times 5$ μm$^3$), the very limited amounts of Mg oxides are not thought to play a role in the observed chemical reactions between nitrogen and magnesium. Indeed, sample characterization by Raman spectroscopy and X-ray diffraction never revealed vibrational modes or diffraction peaks that could be attributed to a solid other than a Mg-N compound.

**Raman spectroscopy**. Sample characterization was achieved in part by confocal Raman spectroscopy measurements performed with a LabRam spectrometer equipped with a ×50 Olympus objective. Sample excitation was accomplished using a continuous He-Ne laser (632.8 nm line) with a focused laser spot of about 2 μm in diameter. The

Stokes Raman signal was collected in a backscattering geometry by a CCD coupled to an 1800 l/mm grating, allowing a spectral resolution of approximately $2\,cm^{-1}$. At ambient pressure, after the release of gaseous molecular nitrogen, the full power Raman laser—4.6 mW incident on the DAC—resulted in the decomposition of the $\alpha$-$Mg_2N_4$ crystallites; the irradiated spots becoming dark and no longer displaying the $\alpha$-$Mg_2N_4$ vibrational modes. To avoid this, the laser power was reduced by employing neutral filters by a factor of approximately 6 (0.75 mW on the DAC).

**X-ray diffraction**. The X-ray diffraction studies were performed at the P02.2 beamline ($\lambda = 0.2901$ Å) at PETRA III. In order to determine the sample position on which the single crystal X-ray diffraction acquisition is obtained, a full X-ray diffraction mapping of the experimental cavity is performed. The sample position displaying the most single crystal reflections belonging to the phase of interest is chosen for the collection, in step-scans of 0.5° from $-36°$ to $+36°$ $\omega$, of the single crystal X-ray diffraction data. The CrysAlis$^{Pro}$ software[31] is utilized for the single crystal data analysis. The analysis procedure includes the peak search, the removal of the diamond anvils' parasitic reflections, finding reflections belonging to a unique single crystal, the unit cell determination and the data integration. The crystal structures are then solved and refined within the JANA2006 software[32]. The procedure for single crystal X-ray diffraction data acquisition and analysis was previously demonstrated and successfully employed[13,14,33,34]. Powder X-ray diffraction was also performed to verify the chemical homogeneity of the samples. The powder X-ray data was integrated with Dioptas[35] and analyzed with the XRDA software[36]. Le Bail refinements employing a powder X-ray diffraction pattern was accomplished with the FullProf software[37].

**Laser-heating**. The double-sided sample laser-heating was performed at our home laboratory at the Bayreuth Geoinstitut using two YAG lasers. Pure magnesium, which is metallic, was employed as the YAG laser absorber. Temperatures were accurately determined from the sample's blackbody radiation[38]. Samples were heated between 10 and 15 min and moved under the YAG beams to evenly heat the whole magnesium piece.

**Atomistic modeling**. Density functional theory (DFT) calculations have been performed using the CASTEP code[39]. The code is an implementation of Kohn-Sham DFT based on a plane wave basis set in conjunction with pseudopotentials. The plane-wave basis set is unbiased (as it is not atom-centered) and does not suffer from the problem of basis-set superposition error unlike atom-centered basis sets. It also makes converged results straightforward to obtain in practice, as the basis set convergence is controlled by a single adjustable parameter, the plane wave cut-off. Pseudopotentials were either norm-conserving or ultrasoft, and were generated using the PBE exchange-correlation functional[40] using the 'on the fly' parameters included in the CASTEP 2019 distribution. These pseudopotentials have been shown to be very accurate and are very well suited for the calculations carried out here[41]. The Brillouin-zone integrals were performed using Monkhorst-Pack grids[42] with spacings between grid points of less than 0.02 Å$^{-1}$. Full geometry optimizations of the unit cell parameters and the internal coordinates were performed until forces were converged to <0.01 eV/Å and the residual stress was <0.02 GPa. Phonon dispersion curves and Raman spectra were computed using linear response theory[43] as implemented in CASTEP[44]. The population analysis was carried out as implemented in CASTEP[45]. Band gaps were obtained from band structure calculations. DFT-GGA calculations tend to systematically underestimate the band gap.

We benchmarked our model calculations by comparison of our results to the experimentally determined ambient conditions structure of $Mg_3N_2$, which adopts the space group $Ia\bar{3}1$ and has a lattice parameter of 9.9528(1) Å[46]. Our calculations gave $a = 10.0127$ Å, thus showing the often observed slight underbinding in GGA calculations. The experimental Raman spectrum[47] is well reproduced, as shown in Supplementary Fig. 3.

## Data availability

The details of the crystal structure investigations may be obtained from the Cambridge Crystallographic Data Centre (CCDC), 12 Union Road, CB2 1EZ Cambridge, United Kingdom (fax: + 44 (0)1223 336033; e-mail: admin@ccdc.cam.ac.uk) on quoting the deposition numbers CSD 1918150–1918155 and CSD 19182019. The data that support the findings of this study are available from the corresponding author upon reasonable request.

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

## Acknowledgements

The authors acknowledge the Deutsches Elektronen-Synchrotron (DESY, PETRA III) for provision of beamtime at the P02.2 beamline. Alexander Kurnosov is greatly thanked for helping with sample loadings employing the high pressure gas loader. D.L. thanks the Alexander von Humboldt Foundation for financial support. N.D. and L.D. thank the Federal Ministry of Education and Research, Germany (BMBF, grants no. 5K16WC1 and no. 05K19WC1) and the Deutsche Forschungsgemeinschaft (DFG projects DU 954–11/1, DU 393–9/2, and DU 393–13/1) for financial support. B.W. gratefully acknowledges funding by the DFG in the framework of the research unit DFG FOR2125 and within projects WI1232.

## Author contributions

D.L., L.D. and N.D. conceptualized the research, D.L., E.K., T.F., M.B. and E.B. participated to the experimental data collection, B.W. performed all theoretical calculations and carried out their analysis, D.L., E.K, T.F. and M.B. conducted the sample laser-heating, D.L. analyzed the experimental results, wrote the original draft and created the figures, D.L., B.W., L.D. and N.D. reviewed and edited the draft.

## Competing interests

The authors declare no competing interests.
