## [Peer Review File · Nature Communications]

Reviewers' comments:

Reviewer #1 (Remarks to the Author):

The authors present their work on synthesis of Mg-N polynitrogen materials via compression of pure Mg and molecular N₂ in laser-heated DAC. Two new compounds emerged above 52 GPa compression: MgN₄ and Mg₂N₄. MgN₄ consisted of infinite nitrogen chains and Mg₂N₄ consisted of tetranitrogen N₄ units. Mg₂N₄ is preserved at ambient conditions while MgN₄ is not.

The results including recovery of Mg₂N₄ at ambient conditions are potentially novel. However, the manuscript does not effectively demonstrate the crystal structure determination neither from experiment nor from theory. Specifically, the Le Bail refinement employed in the work does not allow to determine the positions of the atoms. Therefore, the important question is where the original structures used in their refinement are coming from? In principle, it can come from theoretical predictions, but theory component presented in the current work does not make these predictions. DFT calculations were done only to determine structural information (bond lengths and angles), Raman spectra and electronic properties of already known crystal structures. Therefore, the lack of explanation of the structure determination in the main text makes this work weak and incomplete. The authors should fully discuss all the important details including possible sources of errors in the main text rather than in supplement.

The related issue is the lack of agreement between this work and several theoretical predictions of Mg-N compounds in the three papers cited in the manuscript. In particular, it is stated that MgN₄ possesses Ibam symmetry while the theory predicts first P-1 at low pressure (not ambient however) then a transition to Cmmm at high pressure as well as predicting metastable P2/m phase at ambient conditions in the work on by Yu et al. Neither paper suggest the existence of tetranitrogen structure in crystals with MgN₂ stoichiometry, the main result of this work. Why are the previous calculations, dealing with energetics of various structures, in substantial disagreement with the present experiments? If these structures are metastable, what is the energy difference compared with the lowest one? Are these new structures introduced in this paper dynamically stable?

Some technical issues:

1. The details of calculation of energy content of new crystals including comparison with TNT are missing.

2. DFT calculations are performed with norm-conserving or ultrasoft pseudopotentials, but rationale for using both is not explained. Calculations of thermochemistry requires hard pseudopotential for N, and an appropriate discussion is required.

Reviewer #2 (Remarks to the Author):

This paper describes a careful DAC study of the Mg/N₂ system which resulted in the discovery of one (Authors, correct the title to reflect this) novel polynitrogen salt. Furthermore, the fact that the salt could be recovered at ambient pressure is interesting. The study was carried out competently and is suitable for publication in Nature Comm. The results are mainly of scientific interest and the authors should refrain from exaggerated claims for the potential use of this compound as an energetic material. The Mg₂N₄ salt has a low N-content of only 53.5% and low energy due to the four negative charges. Furthermore, a high-pressure-temperature synthesis is not suitable for scale-up. Out of the more than a dozen known Mg/nitrogen compounds there are at least two stable compounds which can be easily prepared in bulk, Mg(N₅)₂ and Mg(N₃)₂, which have N-contents of 85.2 and 77.6%, respectively and are highly energetic. Therefore, the claims for the potential usefulness of this compound needs to be removed. Not every polynitrogen compound is a highly energetic material. I recommend publication of this paper after minor revision.

Reviewer #3 (Remarks to the Author):

The manuscript by Laniel et al. is an interesting experimental study aimed at synthesizing novel magnesium nitrides from elements at high pressure using laser heating to overcome kinetic barrier of the reaction. The authors managed to synthesize two novel Mg-N binary compounds, MgN₄ and Mg₂N₄, characterized by nitrogen atoms forming polythiazyl-like infinite chains or isolated cis-tetranitrogen (N₄)⁴⁻ anions, respectively. These findings are fundamental from the point of view of inorganic chemistry and important in respect to the quest for novel energetic and environmentally friendly high energy density materials (HEDM). The choice of experimental methods is appropriate and the calculations are in line with the empirical data. To sum up, this is a very good paper and I can gladly recommend it for Nature Communications. However, prior to publication it needs to be improved in terms of clarity of experimental details and interpretation of the results.

The experimental routes and thermodynamic conditions for all performed laser heating experiments are not very clear to me. The authors used four diamond anvil cells and performed laser heating at seven pressure points (28.0, 33.0, 43.4, 52.2, 52.4, 52.7 and 58.5 GPa). However, I do not understand if the laser heating cycles were performed for seven freshly loaded samples of magnesium specks in nitrogen, or some of the samples were laser-heated again, after the first heating cycle and increasing pressure (or pressure increased spontaneously on temperature quenching). While the metallic Mg served as laser absorber, in Supplementary Figures 5 and 6 captions the authors admitted that before the laser heating the sample was composed of C2/m Mg3N2, epsilon-N2 and Mg. Thus, I presume that the sample was heated for the first time in lower pressure to form Mg3N2, and then above 52.4 GPa in the second laser heating cycle. In this respect, the statement found in the Supplementary Fig. 5 caption, namely "Heating is observed to transform the C2/m Mg3N2 into MgN4 and Mg2N4" seems incorrect. Mg3N2 cannot transform nor into MgN4 neither into Mg2N4, since it contains less nitrogen than the high pressure phases, as evidenced by its stoichiometry. Instead it can react with surrounding nitrogen. However, remaining pure Mg can react with nitrogen as well. It would be extremely interesting to verify if two newly synthesized magnesium nitrides were formed from different starting materials (Mg+N2 or Mg3N2+N2). Since the hot spot is localized at the Mg chips, it can be also presumed that the transformed region consisting mainly of Mg3N2 reaches lower temperature during the second laser heating cycle. Unfortunately, since laser heating runs were performed at home laboratory, I doubt that the full mapping of the experimental chamber was carried out before and after the second cycle of laser heating, hence the regions corresponding to different phases cannot be identified.

Nevertheless, the authors should provide all the experimental details in a more clear and detailed fashion, perhaps adding a table to the Supplementary materials. For each fresh loading with Mg and N2 please describe possibly accurately experimental details, i.e. pressure-temperature conditions (e.g. compression, first cycle of laser heating, compression or decompression after laser heating and second cycle if applied). Please provide also details regarding time (10-15 minutes as stated in line 253, but was it the same for all the cycles?) and estimated temperature (above 52.4 GPa it was >1850 K according to information provided in line 69, but what was the heating temperature below this pressure threshold?). Please also estimate temperature uncertainties. Finally, the pressure reported in Supplementary Fig. 5 is 53.5 GPa – should it be 58.5 GPa instead?

There are also minor issues listed below that should be addressed before publication.

Allotrope is a chemical term reserved for different structural modifications of an element (cf. IUPAC Gold Book <https://goldbook.iupac.org/terms/view/A00243>). Please replace "homonuclear polynitrogen allotropes" (line 34) with "polynitrogen species" or "polynitrogen ions". Next sentence should be also corrected: while stabilization of N5+ and N5- ions is a demanding task, inorganic azides (N3-) are stable and well known since the 19th century.

The middle dot is missing in $\text{ReN}_8 \cdot \text{xN}_2$ formula (line 78).

Cl_2^- should be rewritten as Cl^- (line 113).

Le Bail fits are not "structural refinements" (line 247, Supplementary lines 87 and 229), as they do not provide atomic parameters. Please refer to them as "Le Bail refinements" or "Le Bail fits".

Provide missing reference is Supplementary line 100.

Replace "polynitrogen string" (Supplementary line 102) with "polynitrogen chain".

Supplementary Fig. 18 is not cited in the main article, while it illustrates very important crystallographic relation revealed by the authors. This aspect should be addressed properly also in the main manuscript body.

Kamil Filip Dziubek

Response to referees' comments:

To better structure our response to the referees' reports, each of their comments has been copied and our response to each of these comments is written in blue. When changes were made to the manuscript, the original statement from our paper is copied and followed by the updated text.

Reviewer #1 (Remarks to the Author):

The authors present their work on synthesis of Mg-N polynitrogen materials via compression of pure Mg and molecular N₂ in laser-heated DAC. Two new compounds emerged above 52 GPa compression: MgN₄ and Mg₂N₄. MgN₄ consisted of infinite nitrogen chains and Mg₂N₄ consisted of tetranitrogen N₄ units. Mg₂N₄ is preserved at ambient conditions while MgN₄ is not.

The results including recovery of Mg₂N₄ at ambient conditions are potentially novel. However, the manuscript does not effectively demonstrate the crystal structure determination neither from experiment nor from theory. Specifically, the Le Bail refinement employed in the work does not allow to determine the positions of the atoms. Therefore, the important question is where the original structures used in their refinement are coming from? In principle, it can come from theoretical predictions, but theory component presented in the current work does not make these predictions. DFT calculations were done only to determine structural information (bond lengths and angles), Raman spectra and electronic properties of already known crystal structures. Therefore, the lack of explanation of the structure determination in the main text makes this work weak and incomplete. The authors should fully discuss all the important details including possible sources of errors in the main text rather than in supplement.

Regrettably, the reviewer misread the manuscript in parts. In fact, the structures were solved by **single-crystal X-ray diffraction**, as written in the Abstract (line 21) and the main text (lines 72-73, 114-115, 174-175, 176-177 and 204-206) of the revised manuscript. The Methods section—part of the main manuscript—also contains a sub-section “X-ray diffraction” which provides information on the overall **single crystal** data collection method, software employed, and analyses procedures, as well as references to multiples other papers (Refs. 13, 14, 33, 34), in which this procedure is described in detail. Also specified on numerous occasions in the text, the refined single crystal data, allowing to judge the reliability of the solved structures, is available in the Supplementary Materials (Table 2, Mg₃N₂; Table 3, MgN₄; Table 4-Table 7 and Table 9, Mg₂N₄ at various pressures). The full single crystal data were also deposited to the CCDC, with the deposition numbers 1918150-1918155 and 1918209 (please see the newly added “Data availability” section). The Checkcif files related to the aforementioned single crystal data were also made available to the referees by submitting them along with the manuscript. The Le Bail refinement mentioned by the referee is related to powder X-ray diffraction, which was performed to verify the chemical homogeneity of the samples (lines 261-264). The structures obtained from the single crystal X-ray diffraction experiments were afterwards reproduced by our DFT calculations.

As recommended by the referee, we have moved to the main text the crystallographic data previously shown in the Supplementary Materials (Table 1, see below). Table 1 contains the

structural information obtained using single-crystal X-ray diffraction. As written in the manuscript, the full details of the refinement are found in the Supplementary Materials.

Table 1: Crystallographic data, obtained by single-crystal X-diffraction, for the new Mg-N compounds synthesized at the indicated pressures (full crystallographic information is provided in the Supplementary Materials).

	MgN₄	β-Mg₂N₄	α-Mg₂N₄
Pressure (GPa)	58.5	58.5	0.0001 (1 bar)
Space group	Ibam	P2₁/n	P2₁/n
a (Å)	3.5860(13)	7.113(5)	7.5182(9)
b (Å)	7.526(3)	5.828(6)	6.5426(11)
c (Å)	5.1098(17)	8.800(9)	13.4431(19)
β (°)	90	104.00(7)	130.080(17)
V (Å ³)	137.90(9)	354.0(6)	505.95(18)
Fractional atomic coordinates	Mg: (0; 0; 0.25) N1: (0.6584; 0.83344; 0.5)	Mg1: (0.49079; 0.7820; 0.36972) Mg2: (0.87494; 1.0122; 0.65121)	Mg1: (-0.03444; -0.26972; -0.14194) Mg2: (0.19484; -0.24500; -0.43089)
(x ; y ; z)	N2: (0.2928; 0.3059; 0.5)	Mg3: (0.27163; -0.0478; 0.08387) Mg4: (0.73162; 1.0462; 0.26126) N1: (0.9146; 0.7621; 0.4030) N2: (0.5752; 0.2424; 0.4128) N3: (0.5700; 0.4601; 0.3759) N4: (1.0672; 0.6905; 0.5081) N5: (1.0586; 0.7188; 0.6557)	Mg3: (-0.15418; -0.47368; -0.39088) Mg4: (-0.05204; 0.03340; -0.31550) N1: (0.2053; 0.2398; -0.29135) N2: (0.0799; -0.2324; -0.32480) N3: (0.1944; -0.0551; -0.10797) N4: (-0.3561; -0.1649; -0.32605) N5: (-0.0018; -0.5637; -0.18680)

N6: (0.2820; 0.2132; 0.2432)

N6: (0.1442; 0.3256; -0.39668)

N7: (0.4332; 0.1114; 0.3403)

N7: (-0.3255; 0.2339; -0.45463)

N8: (0.7714; 0.8544; 0.4546)

N8: (-0.4397; -0.2489; -0.43838)

The related issue is the lack of agreement between this work and several theoretical predictions of Mg-N compounds in the three papers cited in the manuscript. In particular, it is stated that MgN₄ possesses *Ibam* symmetry while the theory predicts first P-1 at low pressure (not ambient however) then a transition to *Cmmm* at high pressure as well as predicting metastable P2/m phase at ambient conditions in the work on by Yu et al. Neither paper suggest the existence of tetranitrogen structure in crystals with MgN₂ stoichiometry, the main result of this work. Why are the previous calculations, dealing with energetics of various structures, in substantial disagreement with the present experiments? If these structures are metastable, what is the energy difference compared with the lowest one? Are these new structures introduced in this paper dynamically stable?

It is correct that the MgN₄ and Mg₂N₄ structures, which we present in our manuscript, have not been predicted by previous theoretical calculations. This is part of what makes our paper noteworthy. The comparison to previous theoretical calculations is made on page 38 of the revised Supplementary Materials in the section entitled “Comment on previous theoretical predictions of Mg-N solids”. In particular, the *Ibam* MgN₄ phase that we synthesized is crystal chemically closely related to the *Cmmm* structure that was predicted by the calculations (the chains of N-N atoms puckered by Mg atoms are very similar). The obvious differences between the experimental and theoretical structures are the doubled unit cell parameter *c* and the choice of the space group. Fig. 18 of the Supplementary Materials, copied here below, shows the diffraction pattern which illustrates how we get the *c*-parameter double the value of the theoretically predicted one. Indeed, if we draw the reciprocal lattice corresponding to the predicted parameter *c*, then reflections appear at the mid-point of the lattice vector *c** (they are weak but distinct). This means that the *c* value should be two times larger than the predicted one. As for the space group, more than 130 reflections contradict a C-centered lattice assignment, while none contradicts the I-centered lattice. Additionally, as written in lines 274-276 of the revised Supplementary Materials, our own DFT calculations show that the experimentally determined *Ibam* structure has an enthalpy of about ~10 kJ/mol lower than the *Cmmm* structure. Our unambiguous experimental and theoretical findings can therefore be seen as an improvement of earlier work.

We have also checked the calculated enthalpy of our β -P2₁/n Mg₂N₄ structure against that of the P6₃/mcm and the CmcM MgN₂ predicted structures (see revised Supplementary Materials, lines 277-279). We found that β -P2₁/n Mg₂N₄ is energetically competitive with the other predicted structures, i.e. they are all within the uncertainty of our calculations (5 kJ/mol), which explains from a theoretical standpoint why we observe it.

We also provide some insight into why other theoretical structures (such as the MgN₁₀, Mg₅N₄ and Mg₂N₃) were not observed:

“Our results also provide an insight into the predicted Mg-N compounds with the MgN_{10} , Mg_5N_4 and Mg_2N_3 stoichiometry. Since they were not experimentally observed, it can be hypothesized that, like Mg_2N_4 and MgN_4 , they are either stable at a higher pressure than expected, or their Gibbs free energy is simply too high compared to that of the other phases, thus prohibiting their synthesis.”

Supplementary Fig. 1. Experimental 2kl slice of the reciprocal space of MgN_4 at 58.5 GPa. The grid lines correspond to the lattice with a non-double c parameter ($a = 3.5860(13)$ Å, $b = 7.526(3)$ Å, $c = 2.5549(17)$ Å). Encircled in red, the distinctly visible reflections, appearing then in the mid-points of the c^* vector, suggest a double c lattice parameter.

As recommended by the referee, information from the Supplementary Materials was added to the main text in order to clarify some details. The following sentences were added to the paragraph describing the MgN_4 structure (p.4):

“While the MgN_4 compound crystal chemistry matches the theoretical calculations,^{17–19} the measured lattice parameters do not, as the c parameter is double the predicted value and the structure adopts an I -type unit cell, opposite to the calculated C -type (see Supplementary Materials). As shown in Supplementary Fig. 18, by drawing the reciprocal lattice corresponding to the predicted parameter c ($c = 2.5549(17)$ Å), then reflections appear at the mid-point of the lattice vector c^* ; weak but distinctly visible. This means that the c value is actually two times larger ($c = 5.1098(17)$ Å) than the predicted one. The C -centering is contradicted by more than 130 experimentally observed reflections. Our DFT calculations reveal the $Ibam$ β - MgN_4 structure to have an enthalpy of about 10 kJ/mol lower than the predicted $Cmmm$ MgN_4 solid.”

Also, the following sentences were added at the end of p. 7 of the revised manuscript:

“As the β - $P2_1/n$ Mg_2N_4 compound—and its crystal chemistry—had not been predicted by previous theoretical calculations, its enthalpy at 50 GPa was compared to the enthalpies of the $P6_3/mcm$ and the $Cmcm$ MgN_2 structures that had been predicted.^{8,9} We found that β - $P2_1/n$ Mg_2N_4 is energetically competitive with the two predicted structure as all were found to be equal within 5 kJ/mol, which is less than the uncertainty of our calculations.”

Some technical issues:

1. The details of calculation of energy content of new crystals including comparison with TNT are missing.

The TNT data were taken from the literature, as has been stated in the paper.

The calculation we have carried out is a comparison of total energies of α - Mg_2N_4 , Mg_3N_2 and molecular N_2 . A new reference (ref. 28, Zhang et al, PRB **95**, 020103 (2017)), which describes the simple calculations we have performed, has been added to the text of the manuscript.

2. DFT calculations are performed with norm-conserving or ultrasoft pseudopotentials, but rationale for using both is not explained. Calculations of thermochemistry requires hard pseudopotential for N, and an appropriate discussion is required.

The rationale for using a multitude of approaches is to ensure that no systematic errors occur. Also, ultrasoft pseudopotentials allow for fast calculations, so are best suited for initial geometry optimisations of complex, low symmetry compounds. On the other hand, in the DFT implementation we are using, some properties, specifically Raman intensities, can currently only be obtained with norm conserving pseudopotentials. In our specific case, we use the so-called on the fly pseudopotential generators, which allow a seamless switch between the two approaches. The statement that “hard” pseudopotentials are required is imprecise, what is required are **accurate** pseudopotentials. The pseudopotentials we have employed have extensively been tested and have been established to be very accurate, as described in the newly added reference 41.

As recommended by the referee, we have included a relevant comment and reference in the material section of the manuscript:

“These pseudopotentials have been shown to be very accurate and are very well suited for the calculations carried out here.⁴¹”

Reviewer #2 (Remarks to the Author):

This paper describes a careful DAC study of the Mg/N₂ system which resulted in the discovery of one **(Authors, correct the title to reflect this)** novel polynitrogen salt. Furthermore, the fact that the salt could be recovered at ambient pressure is interesting. The study was carried out competently and is suitable for publication in Nature Comm. The results are mainly of scientific interest and the authors should refrain from exaggerated claims for the potential use of this compound as an energetic material. The Mg₂N₄ salt has a low N-content of only 53.5% and low energy due to the four negative charges. Furthermore, a high-pressure-temperature synthesis is not suitable for scale-up. Out of the more than a dozen known Mg/nitrogen compounds there are at least two stable compounds which can be easily prepared in bulk, Mg(N₅)₂ and Mg(N₃)₂, which have N-contents of 85.2 and 77.6%, respectively and are highly energetic. Therefore, the claims for the potential usefulness of this compound needs to be removed. Not every polynitrogen compound is a highly energetic material. I recommend publication of this paper after minor revision.

We would like to thank referee for his/her positive comments. We agree with the referee that the emphasis of the manuscript should be on the discovery of a new polynitrogen entity (N₄⁴⁻) retrievable at ambient conditions rather than the synthesis of a new energetic material since the Mg₂N₄ is not particularly energetic, as rightfully pointed out by the referee. In the revised version of the manuscript, it is written:

“The energy released upon the decomposition of α -Mg₂N₄ into Mg₃N₂ and pure molecular N₂ is calculated to be about 1.9 kJ/g that is about two times less than known for TNT.^{27,28}”, underlining the below-average energetic potential of Mg₂N₄.

We have made further modifications to the text sections that could have magnified the energetic potential of Mg₂N₄:

- The sentence “The new tetranitrogen anion can spark further research for producing novel and more energetic polynitrogen compounds.”

was changed to “The new tetranitrogen anion can spark further research for producing novel and energetic polynitrogen compounds.”

- The sentence “Moreover, these results underline the potential and possibilities that are enabled by high pressure chemistry, along with its applicability to produce compounds relevant at ambient conditions.”

was changed to: “Moreover, these results underline the potential and possibilities that are enabled by high pressure chemistry, along with its applicability to produce compounds compatible with ambient conditions.”

In the new version of the manuscript, the only occurrences of the word “energy” comes up in general statements such as:

- “The synthesis of novel polynitrogen compounds is of fundamental importance due to their potential as environmentally-friendly high energy density materials.”
- “Polynitrogen entities are recognized to be ideal constituents of high energy density materials (HEDM) on account of the tremendous energy released upon the conversion of singly or doubly bonded N-N atoms to triple-bonded molecular nitrogen.¹”

and the only mention of the energetic capabilities of Mg_2N_4 is its aforementioned comparison with TNT.

As suggested by the referee, the concluding sentences alluding to a scale up was removed from the text.

Regarding the Referee’s suggestion to modify the title: we actually report the experimental synthesis of **two**, and not just one, novel Mg-N compounds, namely MgN_4 and Mg_2N_4 . For the MgN_4 compound the composition and topology was indeed previously *theoretically predicted* (see refs. 17-19 of the manuscript), but this compound was *never experimentally synthesized*. Moreover, the unit cell and the space group of the theoretically predicted MgN_4 compound do not reproduce experimental results. These are actually relevant differences. As such, we think that the current title of the manuscript is correct and thus, we have not changed it and hope that the reviewer agrees with this argument. Similarly, we have not been able to find a paper reporting the experimental synthesis of the $\text{Mg}(\text{N}_5)_2$ compound, but only theoretical calculations (X. Zhang *et al.* RSC Adv., 2015, **5**, 21823). If the referee can provide us with the experimental synthesis paper to which he/she refers, we would be grateful.

Reviewer #3 (Remarks to the Author):

The manuscript by Laniel et al. is an interesting experimental study aimed at synthesizing novel magnesium nitrides from elements at high pressure using laser heating to overcome kinetic barrier of the reaction. The authors managed to synthesize two novel Mg-N binary compounds, MgN₄ and Mg₂N₄, characterized by nitrogen atoms forming polythiazyl-like infinite chains or isolated cis-tetranitrogen (N₄)⁴⁻ anions, respectively. These findings are fundamental from the point of view of inorganic chemistry and important in respect to the quest for novel energetic and environmentally friendly high energy density materials (HEDM). The choice of experimental methods is appropriate and the calculations are in line with the empirical data. To sum up, this is a very good paper and I can gladly recommend it for Nature Communications. However, prior to publication it needs to be improved in terms of clarity of experimental details and interpretation of the results.

The authors thank the referee for his thorough review of the manuscript, insightful comments and suggestions.

The experimental routes and thermodynamic conditions for all performed laser heating experiments are not very clear to me. The authors used four diamond anvil cells and performed laser heating at seven pressure points (28.0, 33.0, 43.4, 52.2, 52.4, 52.7 and 58.5 GPa). However, I do not understand if the laser heating cycles were performed for seven freshly loaded samples of magnesium specks in nitrogen, or some of the samples were laser-heated again, after the first heating cycle and increasing pressure (or pressure increased spontaneously on temperature quenching). While the metallic Mg served as laser absorber, in Supplementary Figures 5 and 6 captions the authors admitted that before the laser heating the sample was composed of C₂/m Mg₃N₂, epsilon-N₂ and Mg. Thus, I presume that the sample was heated for the first time in lower pressure to form Mg₃N₂, and then above 52.4 GPa in the second laser heating cycle. **In this respect, the statement found in the Supplementary Fig. 5 caption, namely "Heating is observed to transform the C₂/m Mg₃N₂ into MgN₄ and Mg₂N₄" seems incorrect.** Mg₃N₂ cannot transform nor into MgN₄ neither into Mg₂N₄, since it contains less nitrogen than the high pressure phases, as evidenced by its stoichiometry. Instead it can react with surrounding nitrogen. However, remaining pure Mg can react with nitrogen as well. It would be extremely interesting to verify if two newly synthesized magnesium nitrides were formed from different starting materials (Mg+N₂ or Mg₃N₂+N₂). Since the hot spot is localized at the Mg chips, it can be also presumed that the transformed region consisting mainly of Mg₃N₂ reaches lower temperature during the second laser heating cycle. Unfortunately, since laser heating runs were performed at home laboratory, I doubt that the full mapping of the experimental chamber was carried out before and after the second cycle of laser heating, hence the regions corresponding to different phases cannot be identified.

Nevertheless, the authors should provide all the experimental details in a more clear and detailed fashion, perhaps adding a table to the Supplementary materials. For each fresh loading with Mg and N₂ please describe possibly accurately experimental details, i.e. pressure-temperature conditions (e.g. compression, first cycle of laser heating, compression or decompression after laser heating and second cycle if applied). Please provide also details regarding time (10-15 minutes as stated in line 253, but was it the same for all the cycles?) and estimated temperature (above 52.4 GPa it was

>1850 K according to information provided in line 69, but what was the heating temperature below this pressure threshold?). Please also estimate temperature uncertainties.

As suggested by the referee, a table which summarizes the exact P-T path of our four samples was added to the Supplementary Materials (Supplementary Table 1), and can also be found below. Producing this table made us realize that we had not included the sample pressure after laser-heating. This is contained, along with the pressure value before laser-heating, in the newly produced table. As a side note, the referee is correct that the caption of Supplementary Figure 5 lends the reader to believe that Mg_3N_2 could directly produce MgN_4 or Mg_2N_4 and, while Mg_3N_2 could of course hypothetically decompose into MgN_4+Mg or Mg_2N_4+Mg , in our case we believe that it is rather a reaction with the surrounding N_2 that allows the transformation. To reflect this, the figure's caption was change to:

“Supplementary Fig. 5. Raman spectra of the Mg-N sample before and after laser-heating (LH). Heating allows the C2/m Mg_3N_2 (black) to react with molecular N_2 and produce MgN_4 (green) and Mg_2N_4 (blue). The pressure increases from 52.4 GPa to 53.5 GPa after laser-heating.”

As it can be deduced from the table below, the Mg_2N_4 and MgN_4 compounds can both be produced with $Mg-N_2$ or $Mg_3N_2-N_2$ (with unreacted Mg for absorbing the laser) as precursors. As inferred by the referee, we do not have a good grasp, however, if either precursors lead to the preferential formation of Mg_2N_4 or MgN_4 since we indeed have no proper maps of the samples.

The time duration of the laser-heating is not included in the table as this value is hard to determine. Since the focussed laser-beam is smaller than the size of our Mg pieces, we are constantly moving around the sample rather than staying still in one position for a few minutes. As such, the laser-heating time spent on each area of a Mg piece varies. Typically, shorter amounts of time are spent on the edges of the Mg flake as it reacts much more easily due to the greater surface area in contact with nitrogen. Thus, the most reasonable value we can provide is the 10-15 min estimate.

Table 1: Summary of each sample's pressure-temperature path, along with synthesized phases identified upon laser-heating.

Sample Number	Pressure (GPa) before laser-heating	Pressure (GPa) after laser-heating	Measured temperature range (K)	Synthesized phases
1	28.0*	28.3	2300(200)	Mg_3N_2
	58.1	58.5	1850(200)	$Mg_2N_4 + MgN_4$
2	33.1*	32.8	1900(200)	Mg_3N_2
	43.4	43.7	2000(200)	Mg_3N_2

	52.4	53.5	2300(200)	Mg ₂ N ₄ + MgN ₄
3	52.2*	52.3	2500(200)	Mg ₂ N ₄ + MgN ₄
4	52.7*	54.2	2000(200)	Mg ₂ N ₄ + MgN ₄

*indicates the first pressure at which the sample was laser-heated. Subsequent laser-heating on samples 1 and 2 were performed on the same Mg piece that was previously heated. Between laser-heating pressure steps, the sample was always further compressed. Each sample was decompressed after being laser-heating at their maximum pressure value.

Finally, the pressure reported in Supplementary Fig. 5 is 53.5 GPa – should it be 58.5 GPa instead?

The referee is correct, there is an error in Supplementary Fig. 5. The error is in the caption: the laser-heating pressure was 52.4 GPa, and after laser-heating the pressure jumped to 53.5 GPa, hence the pressure previously written in the figure itself. The pressure value in the caption was changed from 53.5 GPa down to 52.4 GPa. The new caption can be found above in response to the comments of Referee 1.

We would also like to point out that the statement of a pressure accuracy of 0.03 GPa with a ruby ball was removed from the “Methods” section, as it was found to be inexact.

There are also minor issues listed below that should be addressed before publication.

Allotrope is a chemical term reserved for different structural modifications of an element (cf. IUPAC Gold Book <https://goldbook.iupac.org/terms/view/A00243>). Please replace "homonuclear polynitrogen allotropes" (line 34) with "polynitrogen species" or "polynitrogen ions".

We thank the referee for this remark. The word “allotropes” was replaced with “species”.

Next sentence should be also corrected: while stabilization of N₅⁺ and N₅⁻ ions is a demanding task, inorganic azides (N₃⁻) are stable and well known since the 19th century.

The sentence was modified from:

“The stabilization and isolation of these ions is typically achieved in salts with a complex, multi-elements, counter ion.³⁻⁸”

To

“In particular, the stabilization and isolation of the N₅⁺ and N₅⁻ ions is arduous and typically achieved in salts with a complex, multi-elements, counter ion.³⁻⁸”

The middle dot is missing in $\text{ReN}_8 \cdot x\text{N}_2$ formula (line 78).

The dot was added to the formula $\text{ReN}_8 \cdot x\text{N}_2$.

Cl_2^- should be rewritten as Cl^- (line 113).

Cl_2^- was replaced with Cl^- .

Le Bail fits are not "structural refinements" (line 247, Supplementary lines 87 and 229), as they do not provide atomic parameters. Please refer to them as "Le Bail refinements" or "Le Bail fits".

We agree with the referee. The term "structural refinements" was replaced with "Le Bail refinements", as suggested, in both the main manuscript and the Supplementary Materials.

Provide missing reference is Supplementary line 100.

The reference (Fig. 1 of the manuscript) was correctly entered.

Replace "polynitrogen string" (Supplementary line 102) with "polynitrogen chain".

The word "string" was replaced by "chain".

Supplementary Fig. 18 is not cited in the main article, while it illustrates very important crystallographic relation revealed by the authors. This aspect should be addressed properly also in the main manuscript body.

As recommended by the referee, this information was added to the main manuscript along with a reference to Supplementary Fig. 18. The following sentences were added to the paragraph describing the MgN_4 structure:

"While the MgN_4 compound crystal chemistry matches the theoretical calculations,¹⁷⁻¹⁹ the measured lattice parameters do not, as the c parameter is double the predicted value and the structure adopts an I -type unit cell, opposite to the calculated C -type (see Supplementary Materials). As shown in Supplementary Fig. 18, by drawing the reciprocal lattice corresponding to the *predicted* parameter c ($c = 2.5549(17) \text{ \AA}$), then reflections appear at the mid-point of the lattice vector c^* ; weak but distinctly visible. This means that the c value is actually two times larger ($c = 5.1098(17) \text{ \AA}$) than the predicted one. The C -centering is contradicted by more than 130 experimentally observed reflections. Our DFT calculations reveal the *Ibam* β - MgN_4 structure to have an enthalpy of about 10 kJ/mol lower than the predicted $Cmmm$ MgN_4 solid."

REVIEWERS' COMMENTS:

Reviewer #1 (Remarks to the Author):

The authors adequately responded to reviewers' comments, therefore, manuscript can be published as is.

Reviewer #3 (Remarks to the Author):

I am satisfied that the authors have adequately addressed all my suggestions and I consider the paper ready for publication in Nature Communications. I have only one final remark: as the listings of fractional atomic coordinates have been deposited to the CCDC for all determined crystallographic structures, including Table 1 with atomic coordinates in the main text is, in my opinion, redundant. Apparently, the authors have decided to apply this change due to the remark by the Reviewer #1, who clearly overlooked the results of single crystal X-ray diffraction analysis in her/his first report. Therefore, I would leave the decision on including atomic coordinates in Table 1 up to the Editor.

Kamil Filip Dziubek

Answers to the referees' comments:

We thank the two referees for their positive response.

To better structure our response, each of the comments have been copied and our response to each of these is written in blue. When changes were made to the manuscript, the updated text is written.

REVIEWERS' COMMENTS:

Reviewer #1 (Remarks to the Author):

The authors adequately responded to reviewers' comments, therefore, manuscript can be published as is.

We thank the referee for his comments.

Reviewer #3 (Remarks to the Author):

I am satisfied that the authors have adequately addressed all my suggestions and I consider the paper ready for publication in Nature Communications. I have only one final remark: as the listings of fractional atomic coordinates have been deposited to the CCDC for all determined crystallographic structures, including Table 1 with atomic coordinates in the main text is, in my opinion, redundant. Apparently, the authors have decided to apply this change due to the remark by the Reviewer #1, who clearly overlooked the results of single crystal X-ray diffraction analysis in her/his first report. Therefore, I would leave the decision on including atomic coordinates in Table 1 up to the Editor.

Kamil Filip Dziubek

We thank the referee for his comments. We decided to keep the Table 1 in the main manuscript since we find it convenient for readers to have all the main results readily accessible the main manuscript itself.